# Estimating the timing of multiple admixture events using 3-locus linkage disequilibrium

**Mason Liang**[ID][1⊙], **Mikhail Shishkin**[ID][2⊙], **Anastasia Mikhailova**[ID][2], **Vladimir Shchur**[ID][2]*, **Rasmus Nielsen**[ID][1,3]*

**1** Department of Integrative Biology, University of California, Berkeley, California, United States of America, **2** International laboratory of statistical and computational genomics, HSE University, Moscow, Russian Federation, **3** Globe Institute, University of Copenhagen, Copenhagen, Denmark

⊙ These authors contributed equally to this work.
* vshchur@hse.ru (VS); rasmus_nielsen@berkeley.edu (RN)

## Abstract

Estimating admixture histories is crucial for understanding the genetic diversity we see in present-day populations. Allele frequency or phylogeny-based methods are excellent for inferring the existence of admixture or its proportions. However, to estimate admixture times, spatial information from admixed chromosomes of local ancestry or the decay of admixture linkage disequilibrium (ALD) is used. One popular method, implemented in the programs ALDER and ROLLOFF, uses two-locus ALD to infer the time of a single admixture event, but is only able to estimate the time of the most recent admixture event based on this summary statistic. To address this limitation, we derive analytical expressions for the expected ALD in a three-locus system and provide a new statistical method based on these results that is able to resolve more complicated admixture histories. Using simulations, we evaluate the performance of this method on a range of different admixture histories. As an example, we apply the method to the Colombian and Mexican samples from the 1000 Genomes project. The implementation of our method is available at https://github.com/Genomics-HSE/LaNeta.

**Data Availability Statement:** Software is openly available in the repository https://github.com/Genomics-HSE/LaNeta. Data sharing is not applicable as this study only analyses previously published and openly available data. The analysis

## Author summary

We establish a theoretical framework to model 3-locus admixture linkage disequilibrium of an admixed population taking into account the effects of genetic drift, migration and recombination. The theory is used to develop a method for estimating the times of multiple admixtures events. We demonstrate the accuracy of the method on simulated data and we apply it to previously published data from Mexican and Colombian populations to explore the complex history of American populations in the post-Colombian period.

## Introduction

There are many methods for inferring the presence of admixture, e.g. methods using simple summary statistics detecting deviations from phylogenetic symmetry [1–3] and methods

uses genomic data from Yoruba, Colombian and Mexican populations. Whole-genome sequence data are available from the third phase of 1000 Genome Project (http://ftp.1000genomes.ebi.ac.uk/vol1/ftp/release/20130502/). All filters are included in LaNeta in the file prepararion.sh (https://github.com/Genomics-HSE/LaNeta/blob/main/utilites/preparation.sh).

**Funding:** MS and VS worked on this paper within the framework of the HSE University Basic Research Program (hse.ru). AM was supported by the grant RFBR 20-29-01028 (rfbr.ru). RN was supported by NIH grant R01GM138634 (nih.gov). The funders had no role in study design, data collection and analysis, decision to publish, or preparation of the manuscript.

**Competing interests:** The authors have declared that no competing interests exist.

estimating admixture proportions using programs such as Structure [4], Admixture [5] or RFmix [6]. There has also been substantial research on estimating admixture times. Some approaches are based on inferring admixture tract length distributions, such as [7–12]. Over time, recombination is expected to decrease the average lengths of admixture tracts. The length distribution of admixture tracts is therefore informative about the time since admixture. Much of the theory relating to tracts lengths is based on Fisher's famous theory of junctions [13] and subsequent work, such as [14–23]. For example, [24] first discussed the length distribution of tracts descended from a single ancestor. These results informed later analyses of admixture tract length distribution, such as references [7–9]. Gravel [8] also implemented the software program TRACTS, which estimates admixture histories by fitting the tract length distribution, obtained by local ancestry inference, to a exponential approximation.

Another approach, which we will follow in this paper, is based on the decay of admixture linkage disequilibrium (ALD). Linkage disequilibrium exists in any natural population due to mutation and genetic drift. However, in well-mixed and genetically isolated populations with recombination it usually decays quite rapidly at a genomic scale. For example, in many human populations linkage disequilibrium decays to approx. zero in less than 1 Mb. However, admixture tracts introduced into a population in an admixture event generates ALD over much longer distances, even if the amount of LD in the source populations is negligible. After a single admixture event, linkage disequilibrium in the admixed population will then gradually decrease in the subsequent generations as a result of recombination. It is, therefore, possible to make inferences about the admixture history of a population from the patterns of LD present in the population. This insight was first used in the program ROLLOFF [25] and was later extended by ALDER [26]. These two methods use the fact that if an admixed population takes in no additional migrants after the founding generation, the ALD present in the population is expected to decay approximately exponentially as a function of distance. The rate constant of this exponential decay is proportional to the age of the founding admixture pulse and can be used as an estimator. ROLLOFF and ALDER are well suited for inferring the time of the admixture event when the admixture history of the population can be approximated as a single pulse. However, in many realistic scenarios the admixture histories involve multiple pulses. Prominent examples in humans include Native American admixture in Rapa Nui [27] or admixed population groups in the Americas [28]. In these instances the expected decay of LD will become a mixture of exponentials. Existing dating method based on ALD can usually only infer the date of the most recent migration wave [25], or reject the hypothesis of a single pulse admixture [26].

ROLLOFF and ALDER use the information contained in pairs of sites by examining the two-locus linkage disequilibrium between them. Here we extend the theory underlying the methods in ROLLOFF and ADLER to three loci by considering three-locus LD. There are two ways of measuring the linkage between $n$ loci. Bennett [29] defines $n$-locus linkage in a way that maintains a geometric decrease of LD each generation as a result of recombination, which is an important property of two-locus linkage disequilibrium. Slatkin [30] defines $n$-locus LD to be the $n$-way covariance, analogously to the property of two locus LD as the covariance in allele frequency between pairs of loci. For two and three loci, these two definitions coincide, but for four or more loci, they do not. Another method GLOBETROTTER [31] uses a copying model in a way similar to ROLLOFF. First, shared haplotype chunks are inferred with CHROMOPAINTER [32], then a mixture of exponentials is used to fit the coancestry curve.

In this paper, we will use Bennett and Slatkin's definition of three-locus LD to examine the decay of ALD for three sites as a function of the genetic distance between them. We derive an equation that describes the decay of three-locus LD under an admixture history with multiple waves of migration from two source populations. We then compare the results of coalescent

simulations to this equation, and develop some guidelines for when admixture histories more complex than a single pulse can be resolved using ALD. Finally, we apply our method to the Colombian and Mexican samples in the 1000 Genomes data set, using the Yoruba samples as a reference. Fitting a two-pulse model to data, we estimate admixture histories for the two populations which are qualitatively consistent with the results reported in [28].

## Description of the method

### Model

We use a random union of gametes admixture model as described in [33], which is an extension of the mechanistic admixture model formulated by [34]. In this model, two or more source populations contribute migrants to form an admixed population consisting of $2N$ haploid individuals. Each generation in the admixed population is formed through the recombination of randomly selected individuals from the previous generation, with some individuals potentially replaced by migrants from the source populations. For simplicity, we consider a model with only two source populations. Furthermore, the first source population only contributes migrants in the founding generation, $T$. The second source population contributes migrants in the founding generation and possibly in one or more generations thereafter. In generation $i$, for $i = T − 1, \ldots, 0$ (before the present), a fraction $m_i$ of the admixed population is replaced by individuals from the second source population.

**Linkage disequilibrium and local ancestry.** ROLLOFF and ALDER use the standard two-locus measure of LD between a SNP at positions $x$ and another SNP at position $y$, which is a genetic distance $d$ to the right,

$$D_2(d) = \text{cov}(H_x, H_y), \tag{1}$$

where $H_x$ and $H_y$ represent the haplotype or genotypes of an admixed chromosome at positions $x$ and $y$. In the case of haplotype data, $H_{i,x} = 1$ if the $i$th sample is carrying the derived allele at the SNP at position $x$, and is otherwise 0. Alternatively, for genotype data, $H_{i,x}$ take on values from $\{0, 1/2, 1\}$ depending on the number of copies of the derived allele the $i$th sample is carrying at SNP position $x$. We consider an additional site at position $z$, which is located a further genetic distance $d'$ to the right of $y$. The three-loci LD, as defined by [29] and [30], is given by

$$D_3(d, d') = \text{cov}(H_x, H_y, H_z) = \mathbb{E}[(H_x − \mathbb{E}H_x)(H_y − \mathbb{E}H_y)(H_z − \mathbb{E}H_z)]. \tag{2}$$

The LD in an admixed population depends on the genetic differentiation between the source populations and and its admixture history. Let $A_x$ represent the local ancestry at position $x$, with $A_x = 0$ if $x$ is inherited from an ancestor in the second source population (the one which contributed in two admixture events), and $A_x = 1$ if $x$ is inherited from the first source population (the one which contributed in a single admixture event). We can compute $D_3$ in terms of the three-point covariance function of $A_x$ and so separate out the effects of allele frequencies and local ancestry. Consider the conditional expectation $\mathbb{E}(H_x | A_x) = g_x + \delta_x A_x$, where $g_x$ is the allele frequency of locus $x$ in the second source population and $\delta_x = f_x − g_x$ is the difference of the allele frequencies of locus $x$ in the two source populations. We now make the assumption that the allele frequencies in the source populations are known and fixed. Our goal is to prove that

$$D_3(d, d') = \text{cov}(H_x, H_y, H_z) = \delta_x \delta_y \delta_z \text{cov}(A_x, A_y, A_z) \tag{3}$$

By taking expectation, we obtain

$$\mathbb{E}(H_x) = \mathbb{E}(\mathbb{E}(H_x|A_x)) = g_x + \delta_x \mathbb{E}(A_x).$$

Consider an arbitrary number $N$ of sites $S_1, \ldots, S_N$. We assume that $\mathbb{E}(H_{S_i}|A_{S_1}, \ldots, A_{S_N}) \sim \mathbb{E}(H_{S_i}|A_{S_i})$ for any i. Then we have

$$\mathbb{E}\left[\prod_{i=1}^{N}H_{S_i}\right] = \mathbb{E}\left[\mathbb{E}\left[\left(\prod_{i=1}^{N}H_{S_i}\right)\middle|A_{S_0}, \ldots, A_{S_N}\right]\right]$$

$$= \mathbb{E}\left[P\left(H_{S_1} = 1, \ldots, H_{S_N} = 1|A_{S_0}, \ldots, A_{S_N}\right)\right] \quad (4)$$

$$= \mathbb{E}\left[\prod_{i=1}^{N}P(H_{S_i} = 1|A_{S_i})\right] = \mathbb{E}\left[\prod_{i=1}^{N}(f_{S_i} + \delta_{S_i}A_{S_i})\right].$$

Hence, we conclude that

$$\mathrm{cov}(H_{S_1}, \ldots, H_{S_N}) = \mathrm{cov}(g_{S_1} + \delta_{S_1}A_{S_1}, \ldots, g_{S_N} + \delta_{S_N}A_{S_N})$$

$$= \mathrm{cov}(A_{S_1}, \ldots, A_{S_N})\prod_{i=1}^{N}\delta_{S_i}. \quad (5)$$

In particular, we obtain Eq 3 with $N = 3$.

**Local ancestry covariance functions.** From the above section we see that we can describe the three-point admixture LD in terms of covariances of local ancestry in the three points. We now expand the covariance in Eq 2 into its component expectations to get

$$\mathrm{cov}(A_x, A_y, A_z) = \mathbb{E}[A_xA_yA_z] - \mathbb{E}[A_xA_y]\mathbb{E}[A_z]$$
$$-\mathbb{E}[A_xA_z]\mathbb{E}[A_y] - \mathbb{E}[A_yA_z]\mathbb{E}[A_x] + 2\mathbb{E}[A_x]\mathbb{E}[A_y]\mathbb{E}[A_z].$$

Each one of these expectations on the right-hand side is the probability that one or more sites is inherited from an ancestor from the first source population. We organize these products of probabilities in a column vector:

$$\mathbf{v}_3 = \begin{pmatrix} \mathbb{P}\{A_x = A_y = A_z = 1\} \\ \mathbb{P}\{A_y = A_z = 1\}\mathbb{P}\{A_x = 1\} \\ \mathbb{P}\{A_x = A_z = 1\}\mathbb{P}\{A_y = 1\} \\ \mathbb{P}\{A_x = A_y = 1\}\mathbb{P}\{A_z = 1\} \\ \mathbb{P}\{A_x = 1\}\mathbb{P}\{A_y = 1\}\mathbb{P}\{A_z = 1\} \end{pmatrix},$$

so that $\mathrm{cov}(A_x, A_y, A_z) = (1, -1, -1, -1, 2)\mathbf{v}_3$. There is one entry in $\mathbf{v}_3$ for each of the five ways in which the three markers at positions $x$, $y$, and $z$ can arranged on one or more chromosomes. In the founding generation $T$, this column vector is given by $\mathbf{v}_{3(T)} = (1 - m_T, (1 - m_T)^2, (1 - m_T)^2, (1 - m_T)^2, (1 - m_T)^3)'$. The probabilities for subsequent generations can be found by left-multiplying drift, recombination, and migration matrices:

$$\mathbf{v}_{3(i)} = \mathbf{D}_i\mathbf{LU}\mathbf{v}_{3(i-1)},$$

The matrices $\mathbf{D}_i$, $\mathbf{L}$, and $\mathbf{U}$ account for the effects of migration, drift, and recombination, respectively. The migration matrix is a diagonal matrix given by

$$\mathbf{D}_i = \mathrm{diag}(1 - m_i, (1 - m_i)^2, (1 - m_i)^2, (1 - m_i)^2, (1 - m_i)^3).$$

Its entries are the probabilities that one, two, or three chromosomes in the admixed population will not be replaced by chromosomes from the second source population in generation $i$. The lower triangular drift matrix

$$\mathbf{L} = \frac{1}{4N^2} \begin{pmatrix} 4N^2 & 0 & 0 & 0 & 0 \\ 2N & 2N(2N - 1) & 0 & 0 & 0 \\ 2N & 0 & 2N(2N - 1) & 0 & 0 \\ 2N & 0 & 0 & 2N(2N - 1) & 0 \\ 1 & 2N - 1 & 2N - 1 & 2N - 1 & (2N - 1)(2N - 2) \end{pmatrix}$$

gives the standard Wright-Fisher drift transition probabilities between the states as a function of the population size $2N$. Finally, the upper triangular recombination matrix is determined by the recombination rates between the three sites:

$$\mathbf{U} = \begin{pmatrix} e^{-d-d'} & (1 - e^{-d})e^{-d'} & (1 - e^{-d})(1 - e^{-d'}) & e^{-d}(1 - e^{-d'}) & 0 \\ 0 & e^{-d'} & 0 & 0 & 1 - e^{-d'} \\ 0 & 0 & 1 - e^{-d} - e^{-d'} + 2e^{-d-d'} & 0 & e^{-d} + e^{-d'} - 2e^{-d-d'} \\ 0 & 0 & 0 & e^{-d} & 1 - e^{-d} \\ 0 & 0 & 0 & 0 & 1 \end{pmatrix}$$

The covariance function is then given by

$$\mathrm{cov}(A_x, A_y, A_z) = (1, -1, -1, -1, 2)(\prod_{i=0}^{T-1} \mathbf{D}_i \mathbf{L} \mathbf{U}) \, \mathbf{v}_{3(0)}. \tag{6}$$

We can obtain an analogous equation for $\mathrm{cov}(A_x, A_y)$, involving the migration, drift, and recombination matrices for two loci:

$$\mathrm{cov}(A_x, A_y) = (1, -1)\left(\prod_{i=0}^{T-1} \mathbf{D}_i \mathbf{L} \mathbf{U}\right) \mathbf{v}_{2(0)}.$$

In some cases, Eq 6 simplifies further. In a one-pulse migration model, in which the admixture proportion in the founding generation is $m_T = M$ and is there after 0, the $\mathbf{D}_i$'s become identity matrices, and we get the closed from expression

$$\mathrm{cov}(A_x, A_y, A_z) = M(1 - M)(1 - 2M)\left(1 - \frac{1}{2N}\right)^T \left(1 - \frac{2}{2N}\right)^T e^{-T(d+d')}.$$

This is because $(1, -1, -1, -1, 2)$ is a left eigenvector of both $\mathbf{L}$ and $\mathbf{U}$, with corresponding eigenvalues $(1 - 1/2N)(1 - 2/2N)$ and $\exp(-d - d')$. Note that when $M = 0$, the covariance function will be identically 0. Another case is a two pulse model in which we ignore the effects of genetic drift. In this model, admixture only occurs $T$ and $T_2$ generations before the present, so that $m_T = M_1$, $m_{T_2} = M_2$, and all other $m_i$'s are 0. Making the substitution $T_1 = T - T_2$, the

right hand side of Eq 6 becomes

$$-(1-M_1)(1-M_2)e^{-T_2(d+d')}[M_2(1-M_1)^2 - 2M_2^2(1-M_1)^2 + M_1(1-2M_1)e^{-T_1(d+d')} \\ - M_1M_2(1-M_1)(e^{-T_1 d} + e^{-T_1 d'} + (1 - e^{-d} - e^{-d'} + 2e^{-d-d'})^{T_1})]. \quad (7)$$

The corresponding expression for the two-point covariance function is given by

$$(1-M_1)(1-M_2)e^{-T_2 d}(M_2 - M_1 M_2 + M_1 e^{-T_1 d}), \quad (8)$$

which is a mixture of two exponentials.

**Weighted linkage disequilibrium.** As [26] noted, we cannot use the LD in the admixed population directly, because the allele frequency differences in the source populations can be of either sign. As in [26], we solve this problem by computing the product of the values of the three-point linkage disequilibrium coefficient with the product of the allele frequency differences. Using Eq 3 we obtain

$$\delta_x \delta_y \delta_z D_3(d, d') = \delta_x^2 \delta_y^2 \delta_z^2 \mathbb{E}[\text{cov}(A_x, A_y, A_z)],$$

because the local ancestry in the admixed sample is independent of the allele frequencies in the admixed population. For inference purposes, we estimate this function by averaging over triples of SNPs which are separated by distances of approximately $d$ and $d'$. The LD term is estimated from the admixed population, while the $\delta$'s are estimated from reference populations which are closely related to the two source populations. We notice that both this approach, as well as the previous approaches (e.g., [26]), do not take genetic drift in the source populations after the time of admixture into account, i.e. there is an assumption of both this method and previous methods that the allele frequencies in the ancestral source populations can be approximated well using the allele frequencies in the extant populations.

We arrange the data from the admixed samples in an $n \times S_n$ matrix $\mathbf{H}$, where $n$ is the number of admixed haplotypes/genotypes, and $S_n$ is the number of markers in the sample. Similarly, we arrange the data from the two source populations into two matrices, $\mathbf{F}$ and $\mathbf{G}$, which are of size $n_1 \times S_n$ and $n_2 \times S_n$, where $n_1$ and $n_2$ are the numbers of samples from each of the source populations. For ease of notation, we assume that the positions are given in units which make the unit interval equal to the desired bin width.

For a given $d$ and $d'$ the SNP triples we use in the estimator for the weighted LD are

$$S[d, d'] = \{x, y, z : d \leq x - y < d + 1 \text{ and } d' \leq y - z < d' + 1\}.$$

Let $h_x$ be empirical allele frequency in the admixed population. An estimator of the weighted three-point linkage disequilibrium coefficient is then

$$\hat{a}[d, d'] = \frac{1}{|S[d, d']|} \sum_{x,y,z \in S[d,d']} \frac{n \sum_{i=1}^{n} \hat{\delta}_x \hat{\delta}_y \hat{\delta}_z (H_{i,x} - h_x)(H_{i,y} - h_y)(H_{i,z} - h_z)}{(n-1)(n-2)},$$

where

$$\hat{\delta}_x = \frac{\sum_{i=1}^{n_1} F_{i,x}}{n_1} - \frac{\sum_{i=1}^{n_2} G_{i,x}}{n_2},$$

and similarly for $\hat{\delta}_y$ and $\hat{\delta}_z$.

## Algorithm

Directly computing $\hat{a}[d, d']$ over the set $d, d' \in \{0, 1, \ldots, P\}^2$ would be cubic in the number of segregating sites. However, by using the fast Fourier Transform (FFT) technique introduced in ALDER [26], we can approximate $\hat{a}$ with an algorithm whose time complexity is instead linear in the number of segregating sites.

First, rearrange $\hat{a}$ to get

$$\hat{a}[d, d'] = \frac{n}{(n-1)(n-2)} \frac{\sum_{i=1}^{n} \sum_{x,y,z \in S[d,d']} \hat{\delta}_x \hat{\delta}_y \hat{\delta}_z (H_{i,x} - h_x)(H_{i,y} - h_y)(H_{i,z} - h_z)}{\sum_{x,y,z \in S[d,d']} 1},$$

and define sequences $b_i[d]$ and $c[d]$ by binning the data and then doubling the length by padding with $P$ zeros,

$$b_i[d] = \begin{cases} \sum_{x:d \leq \lfloor x \rfloor < d+1} \hat{\delta}_x (H_{i,x} - h_x) & : 0 \leq d \leq P \\ 0 & : P < d \leq 2P \end{cases}$$

$$c[d] = \begin{cases} |\{x : d \leq \lfloor x \rfloor < d+1\}| & : 0 \leq d \leq P \\ 0 & : P < d \leq 2P \end{cases}$$

We can approximate $|S[d, d']|$ and the $n$ sums in the numerator of $\hat{a}[d, d']$ in terms of convolutions of these sequences:

$$|S[d, d']| \approx \sum_{w=0}^{P} c[w]c[w+d]c[w+d+d']$$

$$\sum_{x,y,z \in S[d,d']} \hat{\delta}_x \hat{\delta}_y \hat{\delta}_z (H_{i,x} - h_x)(H_{i,y} - h_y)(H_{i,z} - h_z) \approx \sum_{w=0}^{P} b_i[w]b_i[w+d]b_i[w+d+d'].$$

These convolutions can be efficiently computed with an FFT, since under a two-dimensional discrete Fourier transform from $(d, d')$-space to $(j, k)$-space,

$$\sum_{w=0}^{P} b_i[w]b_i[w+d]b_i[w+d+d'] \leftrightarrow B_i[j]\bar{B}_i[k]B_i[k-j],$$

where $B_i$ is the one-dimensional discrete Fourier transform of $b$ and for $j > 0$, $B_i[-j]$ is the $j^{\text{th}}$ to last most element of $B_i$. Summing over $i$ and taking the inverse discrete Fourier transform, we can approximate the discrete Fourier transform of the numerator of $\hat{a}$. We apply the same method to $c$ to approximate the denominator of $\hat{a}$.

The time complexities for the binning and the FFT's are $O(S_n)$ and $O(P^2 \log(P))$. Of these two, the first term will dominate, because $P$, the number of bins, is much smaller than $S_n$, the number of segregating sites.

## Missing source population

When data from only one source population are available, it is still possible to estimate the weighted admixture linkage disequilibrium by estimating the difference in allele frequencies between the two source populations using the allele frequency differences between the available population and the admixed population [26, 35], by way of the following formula

$$h_x = f_x(1 - M) + g_x M,$$

where $M$ is the admixture proportion. For two pulses of admixture with proportions $M_1$ and $M_2$, a similar equation holds

$$h_x = (f_x(1 - M_1) + g_x M_1)(M_2 - 1) + g_x M_2. \tag{9}$$

The allele frequencies in the missing source population can be estimated from this equation by solving for the relevant unknown term (either $f_x$ or $g_x$). This estimator might be noisy for rare variants, so sites with minor allele frequencies of less than 0.05 should be removed (this corresponds to standard filtering practices for real data).

When using only the admixed population itself as a reference population, the method described above will be biased if the same samples are used to estimate both the linkage disequilibrium coefficients and the weights ($\delta_x$, $\delta_y$, and $\delta_z$). We cannot efficiently compute a poly-ache statistics like [26]. At the cost of some power, we instead adopt the approach of [35] and separate the admixed population into two equal-sized groups. We then use one group to estimate the weights, and the other group to estimate linkage disequilibrium coefficients, and vice versa. This gives two unbiased estimates for the numerator of $\hat{a}$, which we then average.

Another challenge with real data, is that the method might be unstable when admixture proportions are not known. For two pulses of admixture, we have four independent parameters $T_1$, $T_2$ and $M1$, $M2$. In order to simplify the problem, one can estimate the total ancestry fractions of the source populations in the admixed population using `ADMIXTURE` [5] with $K = 2$. Assume that $M$ is the ancestry fraction of the source population which admixed two times. Then

$$M = M_1(1 - M_2) + M_2.$$

This equations are closely related to Eq 9. This allows to reduce the number of independent parameters to 3, which simplifies the optimization problem substantially.

**Fitting the two-pulse model.** We fit Eq 7 to the estimates of the weighted LD using non-linear least squares, with two modifications. We added a proportionality constant to account for the expected square allele frequency difference between the source populations. We also subtracted out an affine term in the weighted LD which is due to population substructure [26]. We estimated this by computing the three-way covariance between triples of chromosomes. We use the jackknife to obtain confidence intervals for the resulting estimates by leaving out each chromosome in turn and refitting on the data for the remaining chromosomes.

## Verification and comparison

We used the package msprime [36] to generate two source populations which diverged 4000 generations ago and a coalescent simulation to generate an admixed population from the two source populations according to two-pulse and constant admixture models. We sampled 50 diploid individuals from the admixed and two source populations, each consisting of 20 chromosomes of length 1 Morgan. The effective population size was $2N = 1000$ for the admixed population and two source populations. Using a two pulse model, we varied the migration probabilities and timings for each pulse to examine the accuracy of Eq 7. We also simulated data for a model with a constant rate of admixture each generation, and compared this to the predictions made by Eq 6.

Our implementation uses Python package `cyvcf2` [37] to read VCF files.

## Patterns of 3-locus LD

We first evaluate the accuracy of the equations developed in this paper by comparing the analytical results to simulated data (Figs 1–3). We find there is a generally a close match between

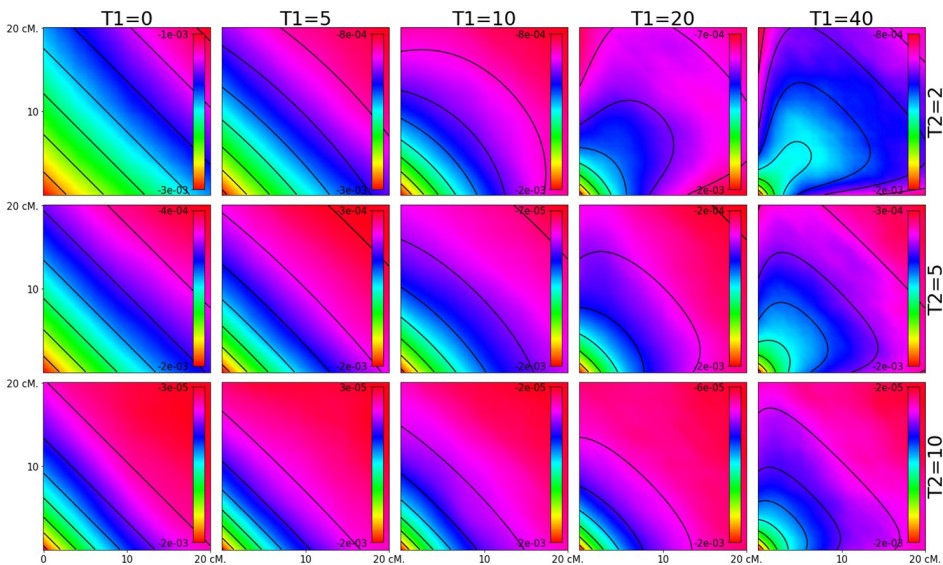

**Fig 1. Predicted weighted LD surfaces from simulations and theory for varying admixture times.** The heat maps are from simulations and the contours are plotted from Eq 7. The two admixture probabilities were fixed at $m_1 = m_2 = .2$ and the the times of the two admixture pulses, $T_1$ and $T_2$, were varied. Each square covers the range 0.5 cM $< d, d' <$ 20 cM. When time of the more recent pulse is greater than half of that of the more ancient pulse, i.e. $2T_1 > T_1 + T_2$, the contours of the resulting weighted LD surface are straight, making it difficult to distinguish from the weighted LD surface produced by a one-pulse admixture scenario.

our equations and the simulated data under both the two-pulse admixture scenarios (Figs 1 and 2) and constant-admixture scenarios (Fig 3). The exception is when the total admixture proportion $M_2 + M_1(1 − M_2)$ is close to 0.5. As the total admixture proportion increases above 0.5, the contours for Eq 2 flip from being concave down to concave up. This transition can be seen by comparing the upper left side of Fig 2 to its lower right. At this threshold, the contours of the estimated weighted LD depend on the actual admixture fractions of the samples, which may differ from the expectation as a result of genetic drift. This mismatch between theory and simulations is most evident in Fig 2, for $m_1 = 0.1$, $m_2 = 0.4$ and $m_1 = 0.2$, $m_2 = 0.4$.

When there is continuous admixture scenario, the shape of the weighted LD surface depends on both the duration and total amount of admixture. When the duration is short, the weighted LD surfaces are indistinguishable from the weighted LD surfaces produced by one pulse of migration. As the duration increases, the contours of the weighted LD surface become more curved. The contours are concave up when the total proportion is greater than 50% and concave down when it is less. When the total proportion is exactly 50%, the amplitude of the weighted LD surface is much smaller than the sampling error.

For two pulse models, the effects of the second pulse of migration only become evident when temporal spacing between the pulses is large enough ($T_1 > T_2$). Otherwise, the resulting weighted LD surface cannot be distinguished from the weighted LD surface produced by one pulse of admixture. As in the case of continuous admixture the concavity of the surface contours is determined by the total admixture proportion.

## Comparison to two-locus LD measures

We compared the simulation results to the two-locus weighted LD calculated by ALDER (Fig 4). The information used in estimating Admixture times in ALDER is the slope of the log-scaled LD curves. Notice (Fig 4) that the slopes are somewhat similar for admixture models

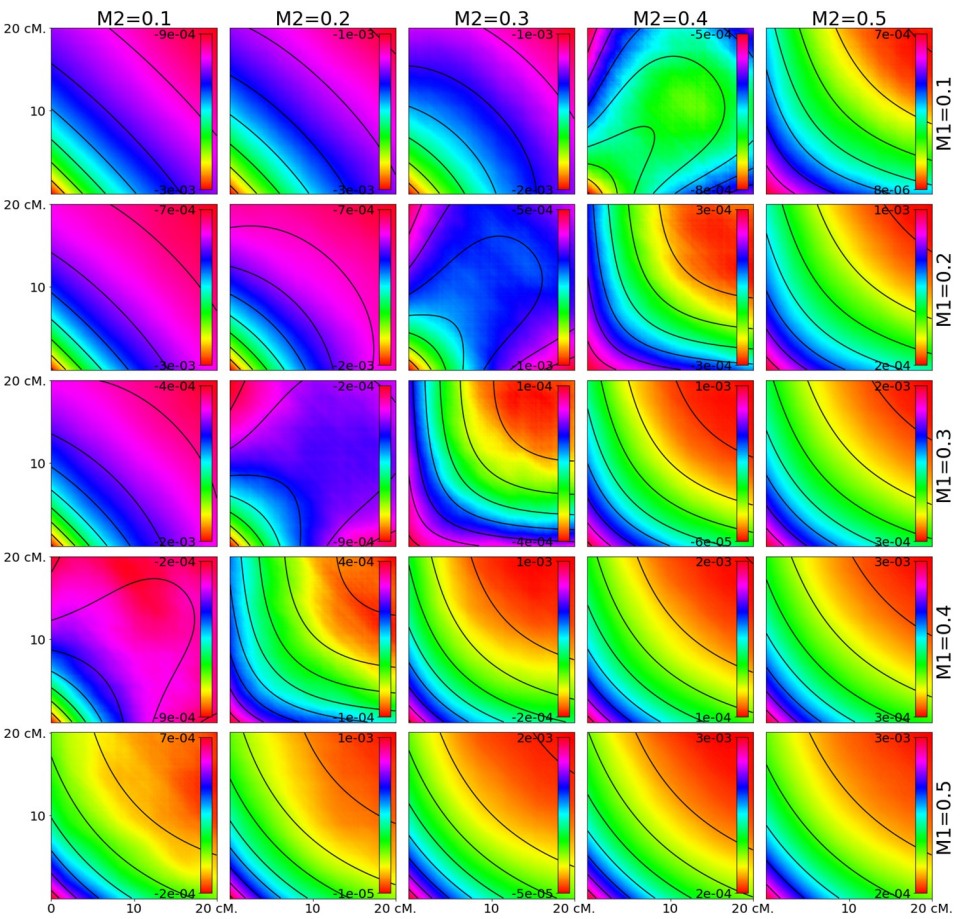

**Fig 2. Predicted weighted LD surfaces from simulations and theory for varying admixture proportions.** The heat maps are from simulations and the contours are plotted from Eq 7. The two admixture times were fixed at 2 and 12 generations ago ($T_1 = 10$ and $T_2 = 2$) while the admixture probabilities were varied. Each square covers the range 0.5 cM $< d$, $d' <$ 20 cM. As the total admixture proportion $m_2 + m_1(1 - m_2)$ increases above 0.5, the contours change to reflecting that the majority contribution of the genetic material now originates from the other population. Weighted LD surfaces for $m_1 > 0.5$ or $m_2 > 0.5$ are not shown, but are qualitatively similar to the surfaces on the lower and rightmost sides.

with identical values of the most recent admixture events ($T_2$). Hence, when two admixture events have occurred, estimation of admixture times tend to get weighted towards the most recent event. Generally, it would be very difficult, based on the shape of the admixture LD decay curve to estimate parameters of a model with more than one admixture event. In contrast, there is a quite clear change in the pattern of three-locus LD as long as the time between the two admixture events is sufficiently large (Fig 1).

## Accuracy of parameter estimates

We next evaluate the utility of the method for estimating admixture times. The qualitative similarities between one pulse and two pulse admixture scenarios seen in the previous simulations under some parameter settings will naturally affect the estimates. As shown in Fig 5, when the spacing between the two pulses is small relative to their age, the median of the estimates of the timing of the second pulse is close to the true value, but the interquartile range is large. Moreover, the best fit often lies on a boundary of the parameter space which is equivalent to a one

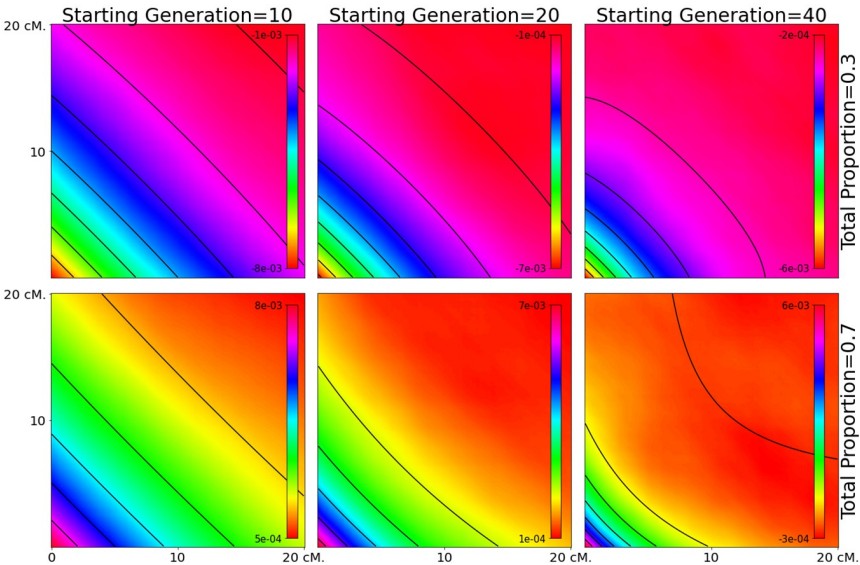

**Fig 3. Weighted LD surfaces produced by constant admixture.** The heat maps are from simulations and the contours from analytical results for a model in which continuous admixture started 10, 20, or 40 generations ago and stopped 5 generations before the present. Each square covers the range 0.5 cM < $d$, $d'$ < 20 cM. We varied the time of the beginning of the admixture and the total admixture probability. The admixture probability for each generation was constant, and chosen so that the total admixture proportion was either 0.3 or 0.7. When the admixture is spread over 5 generations (the leftmost column), the resulting weighted LD surface is similar to a one-pulse weighted LD surface. For longer durations, the weighted LD surfaces are similar to those produced by two pulses of admixture.

pulse admixture model. When the spacing between the pulses is larger, the estimates for the timing of the older pulse become more precise. ALDER estimates a single admixture time (which corresponds to $T_1 = 0$ in our models). There is less variance in this estimate, as it can be explained by a single unknown parameter ($T_2$) compared to three free parameters estimated in our method ($T_1$, $T_2$ and admixture proportion $M_1$).

We evaluated how admixture proportion mis-specification might affect the admixture time estimates. The results are summarised in Fig 6. The timing of the most recent admixture pulse is rather stable to variation in the admixture proportion, while the timing of the older pulse turns out to be quite sensitive to it.

## Applications

To illustrate the utility of the method we computed weighted LD surfaces for Mexican and Colombian samples individuals in the third phase of the 1000 Genomes data set [38]. These samples were previously analyzed for similar purposes by [28]. Our datasets consisted of 64 individuals from Los Angeles and 94 individuals from Medellin, respectively. We used the 104 Yoruba samples as a reference population. We removed indels and SNVs and leave SNPs that only refers to autosomes. (All filters are included in utilites/preparation.sh in https://github.com/Genomics-HSE/LaNeta.) We computed the weighted LD on the genotypes to avoid effects of phasing errors.

For the Mexican samples, [28] found a small but consistent amount of African ancestry, which appeared in the population 15 generations ago, with continuing contributions from European and Native American populations since that date, but no African migration. We fitted a two-pulse model to the Mexican weighted LD surface (Fig 7) with Yoruba as the first source population and the other population being modeled as a missing source population, as

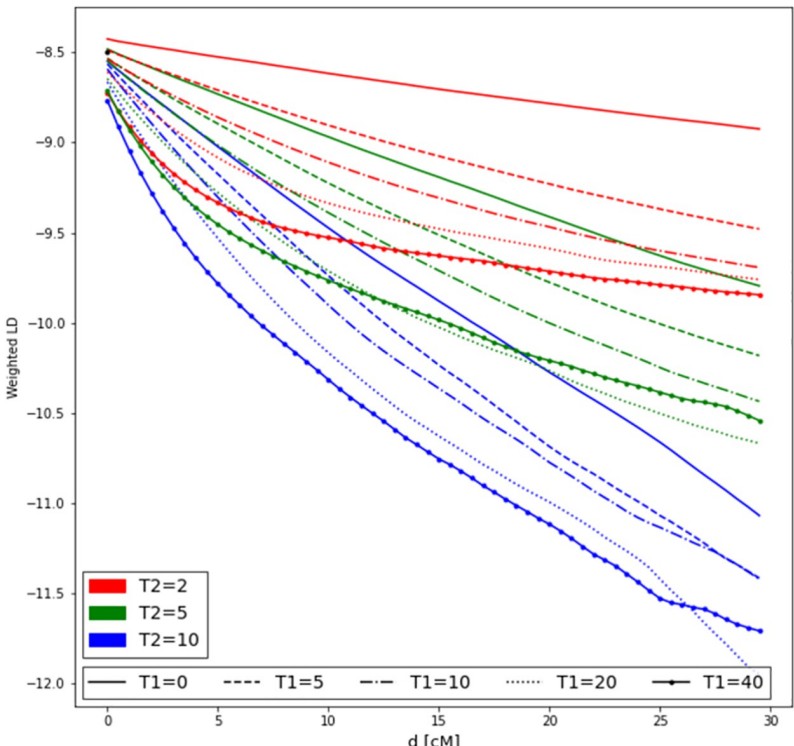

**Fig 4. Two-locus weighted LD with two admixture events and varying pulse times.** Corresponding ALDER curves for two-pulse admixture with varying pulse times. Morgans on *x*-axis and log ALDER scores on *y*-axis. Red lines are $T_2$ = 2, Green lines $T_2$ = 5, and blue lines are $T_2$ = 10.

previously described. The missing source population (non-Yoruban) represents an unknown admixed European and Native American population. This model was chosen to mimic the previous analysis of reference [28] which used a similar model to approximate continued gene-flow from Europeans and Native Americans. Using this set-up we estimated that the two pulses occurred 13.2 ± 1.01 and 7.9 ± 0.99 generations ago. Our results are roughly consistent with those of [28].

The weighted LD surface for the Colombia samples with Yoruba as the first source population is shown in Fig 8. From this, we estimated two pulses of non-Yoruba migration at 14.5 ± 0.74 and 3.7 ± 0.62 generations before the present. [28] inferred two pulses of admixture, corresponding to 13 and 5 generations ago. The weighted LD surface of the Colombian samples has contours which are strongly concave up, in contrast to those of the Mexican samples.

## Discussion

The method presented here is an extension of previously published methods for using weighted two-locus LD to estimate admixture times. The new method uses more information in the data because it compares triples of SNPs instead of pairs. This gives the method the ability to infer admixture histories more complex than a one-pulse model. However, this comes at the price of greater estimation variances. ALDER and ROLLOFF make estimates from just tens of samples, while our method requires hundreds of samples. Part of this difference can be attributed to the fact that ALDER and ROLLOFF make inferences over a smaller class of models, but the main reason arises from the fact that the two-locus methods are estimating second

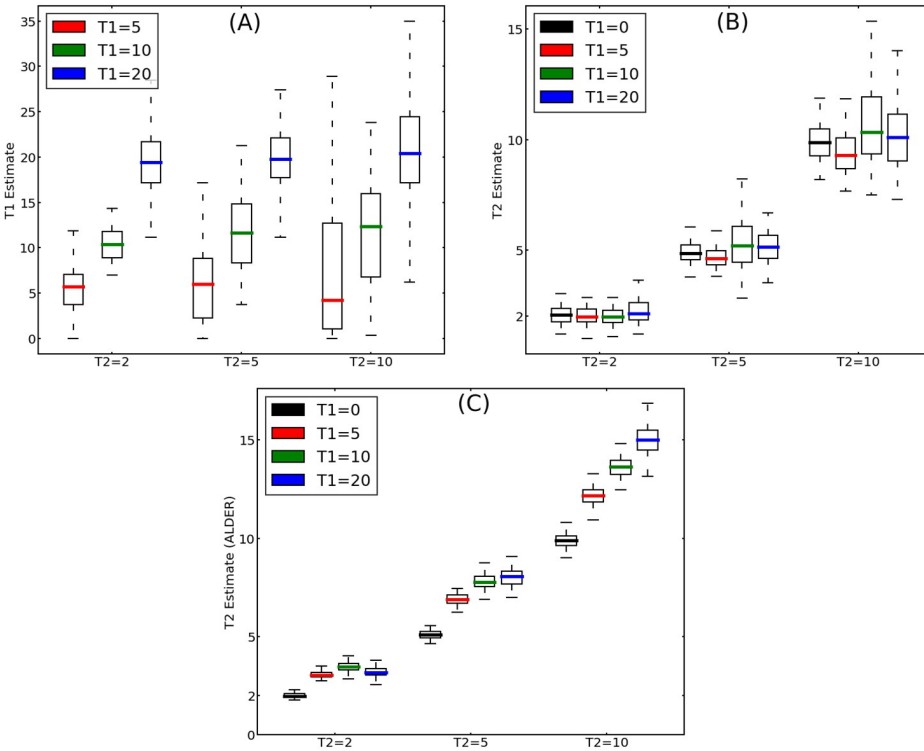

**Fig 5. Accuracy of estimates of $T_1$ (A) and $T_2$ (B), and ALDER estimates of admixture time (C) as a function of other parameters.** Twelve admixture scenarios, $T_1 \in \{0, 5, 10, 20\}$ and $T_2 \in \{2, 5, 10\}$, were simulated 100 times each. The admixture probabilities were fixed at $M_1 = 0.3$ and $M_2 = 0.2$. The colored bars give the medians of estimates for each of these twelve cases, the boxes delimit the interquartile range, and the whiskers extend out to 1.5 times the interquartile range. As the time between the two pulses of admixture increases, the error in the estimates decreases (for this reason we do not include $T_1 = 0$ accuracy estimate, in this case the results become unreasonable). Consistent with the simulations shown in Fig 1, there is limited power to estimate the time of the more ancient admixture pulse when $T_2 > T_1$. ALDER estimates a single admixture time which corresponds to $T_1 = 0$.

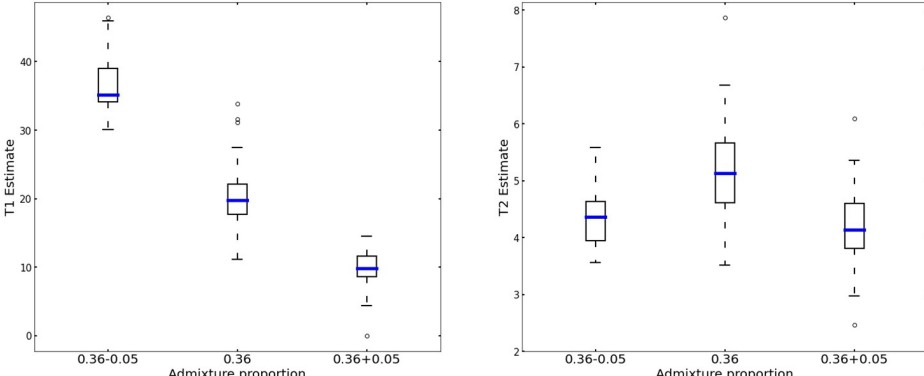

**Fig 6. Effect of admixture proportion misspecification on the estimated values of $T_1$ and $T_2$.** Admixture proportion misspecification has a strong effect on the estimates of time $T_1$ between pulses of admixture. Estimates of the time $T_2$ of the most recent admixture pulse remain stable.

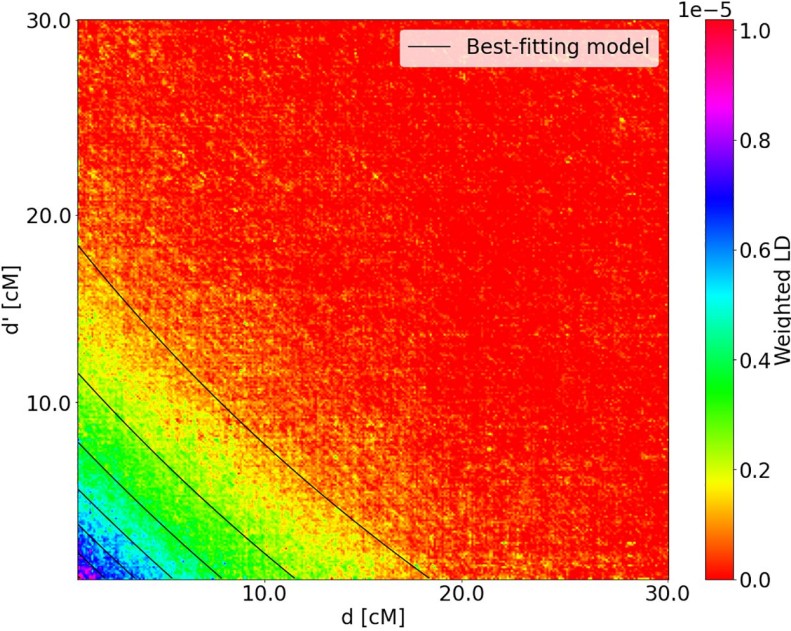

**Fig 7. Weighted LD surface for Mexican samples with Yoruba as the first source population reference.** The model with the best fit is two pulses from the non-Yoruba source population at $T_1 + T_2 = 13.2 \pm 1.01$ and $T_2 = 7.9 \pm 0.99$ generations ago. The weighted LD surface was estimated from real data, the level lines correspond to the best-fitting model inferred by LaNeta method.

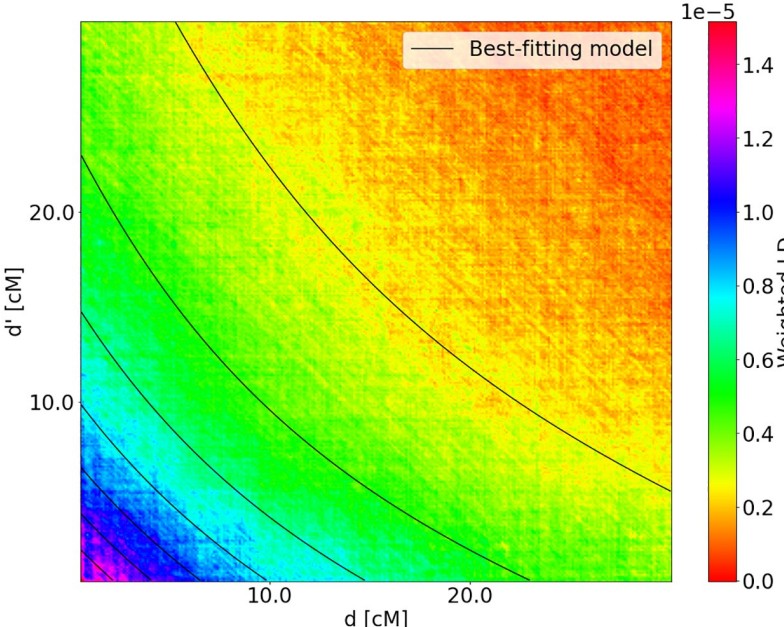

**Fig 8. Weighted LD surface for Colombian samples with Yoruba as the first source population reference.** The two-pulse model that fits best is two pulses of non-Yoruba admixture at $T_1 + T_2 = 14.5 \pm 0.74$ and $T_2 = 3.7 \pm 0.62$ generations ago. The amplitude of this weighted LD surface is approximately ten times larger than that of the Mexican samples. This is a result of larger proportion of Yoruba ancestry in the Colombian samples. The weighted LD surface was estimated from real data, the level lines correspond to the best-fitting model inferred by LaNeta method.

moments of the data, while we are estimating third moments. The variance of these estimates are both inversely proportional to the sample size, but the constants for estimating third moments are larger. As data becomes more readily available, this disadvantage should disappear.

We also note that the theory developed in this paper might be useful for other purposes than estimating admixture times. In particular, it can be used to test hypotheses regarding the spatial distribution of introgressed fragments in the genome, without relying on particular inferences of admixture tracts.

## Acknowledgments

This research was supported in part through computational resources of HPC facilities at HSE University [39].

## Author Contributions

**Conceptualization:** Rasmus Nielsen.

**Data curation:** Anastasia Mikhailova.

**Formal analysis:** Mason Liang, Mikhail Shishkin, Vladimir Shchur, Rasmus Nielsen.

**Methodology:** Mason Liang, Mikhail Shishkin, Vladimir Shchur, Rasmus Nielsen.

**Software:** Mikhail Shishkin.

**Supervision:** Vladimir Shchur, Rasmus Nielsen.

**Validation:** Mason Liang, Mikhail Shishkin.

**Visualization:** Mason Liang, Mikhail Shishkin.

**Writing – original draft:** Mason Liang, Mikhail Shishkin, Vladimir Shchur, Rasmus Nielsen.

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
