## [Decision Letter · Decision Letter 0]

24 Mar 2022

Dear Dr Nielsen,

Thank you very much for submitting your Research Article entitled 'Estimating the timing of multiple admixture events using 3-locus Linkage Disequilibrium' to PLOS Genetics.

The manuscript was fully evaluated at the editorial level and by independent peer reviewers. The reviewers appreciated the attention to an important topic but identified some concerns that we ask you address in a revised manuscript

We therefore ask you to modify the manuscript according to the review recommendations. Your revisions should address the specific points made by each reviewer.

[LINK]

Yours sincerely,

Garrett Hellenthal

Guest Editor

PLOS Genetics

David Balding

Section Editor: Methods

PLOS Genetics

Editor comments:

Thank you for your manuscript. Both reviewers were positive about your paper and believe it is likely worthy of publication in PLoS Genetics after minor amendments. We agree, and have the following points to add:

(1) It would be helpful to make clear in the Introduction that this new technique is designed to detect multiple admixture pulses involving two sources, rather than (e.g.) involving three distinct sources.

(2) Following from this, the 1000 Genomes applications were a bit unclear, given you analyse populations that are admixed from three sources. For example, what do you mean by "with Yoruba as reference"? I think you are assuming two pulses of admixture from a non-African source into an African one for these analyses? If so, it is not clear why this makes more sense than considering two pulses of African admixture. Also, what is the "non-Yoruba" source here -- is it an admixed Native/European source? It would be helpful to explicitly spell out what you used as surrogates for the admixing sources 1 and 2 here.

(3) Your paper figures are not likely to be very intuitive for most readers, including us. What are the lines in Figure 6 and 7, for example? Are these representing the best-fitting model? Most readers are likely to be more interested in plots like Figure 5, where you show estimation accuracy. But why is this only shown for T1 here, and not for T2? Readers might also be interested in what happens when the admixture proportions fixed in your model are inaccurate. While this is mentioned as a potential issue, at what point does it break down?

(4) Detailed theory and applications involving multiple pulses of admixture, or specifically admixture between two sources followed by additional admixture from another source, have been described previously (Hellenthal et al 2014, Science 343:747; Ni et al 2018, Heredity 121:52), yet you do not reference these here. I appreciate your approach is different and has novelty, but it might be worth citing these in the Introduction.

Reviewer's **Comments to the Authors:**

Reviewer #1: The authors provide an interesting exploration of the genomic effects of admixture by leveraging information from three linked sites. I believe the main value of their work stands in the theoretical advancement and demonstration of its match with coalescent simulations, and by itself this part (which I was not able to fully evaluate due to my knowledge gap) is worth of publication unless other reviewers found major issues that I may have overlooked.

I have only minor concerns regarding the practical applications of the work:

1) It is not clear whether the model choice (one/two/continuous waves of admixture) is a decision to be made a-priori by the operator or whether that appears from the results (eg. in Figure 6, should one look at the green portion of the plot to find out which are the best parameters along the Y and X axes?)

2) Although the ALDER output is somewhat used (Fig 4) as a reference for their novel results, it would be desirable to have a more clear benchmarking to assess whether A) the novel method is capable of discriminating between one, two or continuous admixture scenarios and B) by adding the ALDER (or MALDER) performance to figure 5, to what extent is the new approach better/comparable in retrieving admixture dates from the simulated data?

Reviewer #2: In this paper the authors build upon existing theory for estimating the timing of admixture events based on the decay of two-locus admixture linkage disequilibrium (ALD) by developing a framework for a three-locus model. The major advantage of this new model is that it is capable of disentangling the timings of more than one wave of admixture. The authors test their model extensively against simulated data for a number of multi wave admixture scenarios (e.g. different admixture event timings, different admixture proportions and two-pulse vs continuous admixture) to establish how well it performs under a range of conditions. Finally they test their method on real data by modelling two-pulse admixture in Mexican and Columbian data from the 1000 genomes and comparing their results with the findings of Gravel et al. 2013.

I believe this article would be of great interest to readers of PLoS Genetics. The theory developed within is novel, rigorous and extensively tested, and has the potential to improve the study of admixture timing. The authors have carefully outlined the situations in which their model performs best and those in which it loses power which will be of great help to researchers attempting to apply it to their own data. Last but not least they have provided an implementation of their code making the method freely accessible to the research community. For these reasons I would recommend that the article is published once a few minor issues are addressed (see below)

1.) Reference to equations in text:

I noticed two instances where the equation referred to in the text does not seem to match the equation being used in the analysis. I could be entirely mistaken here, but I thought I’d point them out just to be safe.

1.1) References to equation 8: Equation 8 is mentioned twice in the paper as the model used for two-pulse admixture (Line 212; Lines 225-227). However when inspecting this equation in the methods it appears to be a two-locus model rather than a three-locus model (only has a single genetic distance parameter d and no second d’ parameter). Further supporting this interpretation, the line preceding equation 8 (line 134) explicitly states that it is an “expression for the two-point covariance function”.

My question is did you mean to use the two-locus model here (eq 8) or is this supposed to be equation 7? (Eq 7 appears to be the 3-locus form)

1.2.) References to equation 2: From the results section (lines 237-239 page 14) it seems like the model being tested in figures 1 and 2 is a two-pulse admixture model (Eq 7 or eq 8 depending on the answer to the above point). However in the figure legends of figures 1 and 2 (and on line 241) it is stated that the contours in these figures come from equation 2. Is this correct? I don’t see any parameters for admixture timings, or proportions (the variables being explored in the plots) in equation 2 so I am unclear how this would work.

2.) Issues with figures:

2.1.) Figures 1-3:

While the message of these plots regarding how well theory and simulations line up under different admixture scenarios is nicely illustrated, I initially found it difficult to determine what exactly is being plotted in these figures for the following reasons:

i.) No axis labels: The heatmaps lack any x and y axis labels making it unclear at first glance what is being plotted. Please provide axis labels.

ii.) Parameter labels where axis labels expected: The grid labels describing the parameters for a given simulation (e.g. T1=0, T2=10 for figure 1) fall where I would expect the missing x and y axis labels to be (bottom and left sides of a plot) leading to further confusion. Maybe separating these grid labels from the plots using a dividing line, or placing them to the right and top of the plot would help to avoid confusion.

iii.) No scale of heatmap: There is no scale provided for the heatmaps meaning the colours are hard to interpret. Please provide a scale bar.

2.2) Figure 4:

This figure needs a few adjustments to improve legibility I recommend the following tweaks:

i.) Axis labels: Please label the x and y axes in the plot for clarity. The axes are currently described in the figure legend but readability would be increased by labelling the axes directly.

ii.) Plot key: The key in the figure could be cleaned up by reordering it into three columns grouped by T2 values (i.e. First column is all T2=2, second is all T2=5 and third is all T2=10) to improve figure legibility. This way all reds, all greens and all blues are together and the reader will understand that the colour scheme refers to the T2 values intuitively. This will also make the message of the plot clearer.

iii.) T1 values ambiguous: Currently there is no way to distinguish which lines belong to which value of T1 as the colours of each line within a T2 value group are identical. This could easily be solved by using different line types for each value of T1 (i.e. solid lines for T1=10, dashed lines for T1=5 and dotted lines for T1=10) or different shades of the grouping colour (i.e. T1=0 is light, T1=5 is medium, T1=10 is dark).

2.3.) Figures 6-7: Labelling the scale bar here may be helpful to readers.

**Have all data underlying the figures and results presented in the manuscript been provided?**

Reviewer #1: Yes

Reviewer #2: Yes

PLOS authors have the option to publish the peer review history of their article (what does this mean?). If published, this will include your full peer review and any attached files.

Reviewer #1: No

Reviewer #2: **Yes: **Ross Patrick Byrne

---

## [Editor Report · Decision Letter 1]

2 Jun 2022

Dear Dr Nielsen,

We are pleased to inform you that your manuscript entitled "Estimating the timing of multiple admixture events using 3-locus Linkage Disequilibrium" has been editorially accepted for publication in PLOS Genetics. Congratulations!

Thank you for addressing the editor and reviewer concerns.  In preparing the final version for  publication in PLoS Genetics, we ask you to make minor changes to address the following two comments:

1) In Verification and Comparison, please remind readers of the definition of T1 and T2, as this may be missed for those that only quickly peruse the Methods.

2) In Fig 6 state the true dates of T1 and T2, and true admixture proportion here (presumably 0.36 ?)

In addition, before your submission can be formally accepted and sent to production you will need to complete our formatting changes, which you will receive in a follow up email. Please be aware that it may take several days for you to receive this email; during this time no action is required by you. Please note: the accept date on your published article will reflect the date of this provisional acceptance, but your manuscript will not be scheduled for publication until the required changes have been made.

Yours sincerely,

Garrett Hellenthal

Guest Editor

PLOS Genetics

David Balding

Section Editor: Methods

PLOS Genetics

**Data Deposition**

http://datadryad.org/submit?journalID=pgenetics&manu=PGENETICS-D-21-01477R1

**Press Queries**

---

## [Editor Report · Acceptance letter]

12 Jul 2022

PGENETICS-D-21-01477R1 

Estimating the timing of multiple admixture events using 3-locus Linkage Disequilibrium 

Dear Dr Nielsen, 

We are pleased to inform you that your manuscript entitled "Estimating the timing of multiple admixture events using 3-locus Linkage Disequilibrium" has been formally accepted for publication in PLOS Genetics! Your manuscript is now with our production department and you will be notified of the publication date in due course.

With kind regards,

Olena Szabo

PLOS Genetics

On behalf of:
